# Guilt Feelings in Obsessive Compulsive Disorder: An Investigation between Diagnostic Groups

**DOI:** 10.3390/jcm11164673

**Published:** 2022-08-10

**Authors:** Alessandra Mancini, Umberto Granziol, Andrea Gragnani, Giuseppe Femia, Daniele Migliorati, Teresa Cosentino, Olga Ines Luppino, Claudia Perdighe, Angelo Maria Saliani, Katia Tenore, Francesco Mancini

**Affiliations:** 1Schools of Cognitive Psychotherapy (APC-SPC), Viale Castro Pretorio, 116, 00185 Rome, Italy; 2Department of General Psychology, University of Padova, Via VIII Febbraio, 2, 35122 Padova, Italy; 3Dipartimento di Scienze Umane, Guglielmo Marconi University, Via Plinio, 44, 00193 Rome, Italy

**Keywords:** obsessive-compulsive disorder, deontological guilt, depression, interpersonal guilt, guilt

## Abstract

Guilt plays a role in various forms of psychopathology. However, different types of guilt might be involved in different mental disorders. Obsessive-compulsive (OC) patients are prone to a type of guilt in which the violation of an internalized moral norm is necessary and sufficient, whereas data suggest that depression might be linked to more interpersonal types of guilt. However, the extent to which a specific guilt phenomenology is involved in each condition is yet to be determined. Here we assessed the association between different types of guilt and different diagnostic groups. Two clinical samples (33 OCD and 35 non-OCD) filled in the Moral Orientation Guilt Scale (MOGS) along with other OCD and depression measures. Regression was employed to test group differences in the MOGS subscales and to test the influence of MOGS subscales on OCD and depression levels. Results confirm that different types of guilt might be implicated in different psychopathological conditions. Specifically, moral norm violation guilt is more present in OC patients than in other disorders. Depression seems to be associated with different guilt feelings depending on the psychopathological condition, specifically in non-OC patients, with types of guilt involving a “victim”, supporting the accounts viewing interpersonal guilt as involved in the emergence of depressive symptomatology and hyper-altruistic behavior as a vulnerability factor for depression.

## 1. Introduction

Psychopathology and guilt are often intertwined. However, guilt is not a univocal concept, and understanding which type of guilt is involved in different psychopathological conditions is important to improve clinical practice. One way to conceptualize guilt is by considering individual moral orientation. According to this view, there can be guilt feelings related to the deontological moral domain and the altruistic moral domain. In order to experience deontological guilt (DG), the transgression of an interiorized moral norm is necessary and sufficient, while in order to experience altruistic guilt (AG), the failure of an altruistic goal and the presence of a victim are necessary, even if no moral norm has been trespassed [1]. The first empirical tool allowing us to measure different types of guilt according to moral orientation is the 17-items Moral Orientation Guilt Scale (MOGS), recently developed and validated by the authors [2]. The MOGS identifies four different factors:

(i) Moral Norm Violation–(MNV), corresponding to the fear of having outraged an authority and the attempt to prevent guilt by conforming to moral norms (e.g., Item 7: “I feel guilty if do not respect the figures that, to me, represent authority.”). The following vignette aptly represents a possible real-life scenario in which this type of guilt might be experienced (Lara): “when I was 18-year old I was invited to a party with some of my new friends from university. I would really like to go, because I knew that the boy I liked would have been there, but the party was far from my hometown and my father, who was a very strict man, forbid me to use his old car. I than sneaked out at night and stole my father’s car. I went to the party and parked the car in the street in front of my friend’s house. After a few hours, I decided to go back home but when arrived at the parking spot I noticed that the car had a scratch on one of the side doors. I immediately realized what I had done and pictured the image of my father’s severe expression thinking: “I could have been more careful, He’d be right to be angry”.

(ii) Moral Dirtiness–(MODI), corresponding to the propensity to feel morally degraded when feeling guilty and to experience moral disgust towards oneself (e.g., Item 1: “When I feel guilty, I feel dirty inside”). An example of this type of guilt is aptly represented in the following scenario, inspired from a real-life account (Paul): When I was 6 my best friend was my cousin Mark who lived next door. We played together all the times and one day, we decided it might be fun to play pretend doctor. To increase the realism, I suggested we should get naked. While we were playing naked, my mother got in the room and asked: “What are these dirty games!?”. I noticed that she was giving me a very disappointed look, almost disgusted, and I felt like I had lost some worth in the eyes of my mother.

(iii) Empathy corresponds to the propensity to feel guilty for the misfortune of others (e.g., Item 17. “I tend to feel guilty if I do not help who is suffering”). This type of guilt is aptly depicted in the following vignette (Valerie): “One night I went clubbing with a group of friends. We danced all night and had a lot of fun. It was very late when we decided to head back home, walking. Outside it was a chilly winter night and while we were talking, we could see the smoke coming out our mouths. Waiting for a streetlight to become green, I spotted a guy, approximately our age, sitting in the corner of the street, shivering. His clothes were dirty and he had his arms on his head. When the light turned green I hesitated but all my friends kept walking and after a moment I reached them on the other side of the road. The morning after I remembered about that guy in the street. I felt sorry for him and somewhat guilty. I thought that I should have helped him”.

(iv) Harm corresponds to the propensity to feel and prevent guilt resulting from harming others and the tendency to comfort the victim (i.e., Item 15: “If I hurt someone, I feel guilty for the harm I caused”). The following scenario represents a possible antecedent to this type of guilt. “My brother and I were always rivaling. He is just 1 year younger than me and we always did the same sports. When we were 20 we used to do running races. One day, we were running and I was winning, when I accidentally came too close to him. He tripped and fell and I heard a distinct crack. Once I saw him on the floor bleeding, I realized that I accidently had hurt him. In that moment I started to feel very guilty, he was suffering and, even if I didn’t want to hurt him, it was my fault.”

Even if different types of guilt are often experienced at the same time in everyday life [3], it is worth noting that in the first two vignettes, there is no “victim”, while in the last two vignettes, there is no moral transgression. Similarly, items contained in MNV and MODI subscales are focused on moral transgression and self-loathing with no reference to concern for others, while in the Empathy and Harm, items there is always somebody else who is suffering even though no violation of an inner moral rule has occurred (see the Appendix A for MOGS items).

At the social level, guilt feelings serve the evolutionary function of preserving social order and promoting pro-social behavior [4,5]. At the individual level, an enhanced propensity to experience guilt feelings and act upon them (e.g., trying to avoid them) is routinely observed in several psychopathological conditions [6]. The central role of guilt in psychopathology is also well-represented, for example, in the analytic tradition, in which guilt is a necessary consequence of individuation [7]. In order to rise up above conventional morality and become authentic, one must leave “God, mother and the crowd” [8]. Consistently, overinvestment in the attempt to avoid and prevent the burden of guilt leads to a failure in individuation and to mental disorders. Hence, there is a suggestion that the confrontation with guilt is the first task of psychological growth (i.e., psychotherapy) [9]. Here, we propose that the phenomenology of guilt, involved in different mental disorders, varies according to the type of “original sin” that one is afraid to commit. From the clinical point of view, understanding which type of guilt the patient is trying to avoid or, using a more cognitivist vocabulary, towards which goal the patient is sinking his (dysfunctional) efforts, is fundamental to building a meaningful and useful treatment plan. Indeed, intervening in types of guilt that deal with moral transgression and self- moral disgust has different implications than intervening in a more interpersonal type of guilt focused on the misfortune of others or harm caused to another. Namely, the first intervention should address the relationship with moral authority (e.g., God), bringing the patient from a strictly hierarchical conception of it (e.g., the idea that a “sin” is unforgivable and that after the “sin” the relationship with the authority is irreparably compromised) to a more “evangelic” vision where fallibility is intrinsic in human nature and self-forgiveness is possible [10]. The second type of intervention should assess the causal nexuses between the patient’s behavior and the status of the other person, the construction of an overall balance of guilt, and prosocial behaviors performed by the patient in their life taken as a whole, and, ultimately, it should consider intentionality.

The role of guilt in OCD has long been theorized [11] and more recent data confirmed stronger feelings of guilt among OCD patients as compared to matched controls [12]. Specifically, mounting evidence suggests that the guilt feared by obsessive-compulsive (OC) patients is mainly “deontological” and that they are more sensitive to deontology than non-obsessive individuals [13,14]. Consistently, moral decision-making studies have determined that OC patients are more prone to choose in order to respect the “Do not Play God principle” (according to which no one has the right to make a decision that does not respect the limits of their social rank [15]) with respect to healthy controls and patients with anxiety disorders [16,17,18]. Moreover, functional magnetic resonance imaging (fMRI) studies have shown altered processing of DG stimuli in OC patients with respect to control participants. No significant differences were observed between groups when processing AG, angry, or sad stimuli [19]. Relevant to the present work, stimuli-evoking DG consisted of the association between an angry face and sentences such as ‘‘How could I behave that immorally!”, while AG stimuli consisted of the association of a sad face with a sentence such as ‘‘How unfair! I am doing so well, while she/he is so unlucky!’’. Consistently, the activation of the anterior cingulate cortex, the insulae, and the precuneus was observed in OC patients undergoing a symptom provocation task [20,21], suggesting that during symptoms, patients might be experiencing a type of guilt connected to the violation of a moral norm. Most importantly, DG and AG processing is underpinned by two distinct neural pathways. Namely, while DG induction has been associated with the activation of the anterior cingulate cortex, the insula and the precuneus [22], i.e., brain areas that are also implicated in the processing of disgust and self-loathing [23], AG stimuli have been associated with activation in the circuitry implicated in the theory of mind [24]. Importantly, previous studies showed a bidirectional relationship between depression and altruism [25] and that excessive interpersonal guilt might be a precipitating factor contributing to the emergence of internalizing problems, such as anxiety and depression [26]. Finally, empathy-based guilt has been associated with hyper-altruism in MDD [27,28,29].

Thus, different types of guilt might be implicated in different psychopathological conditions. However, the extent to which a specific guilt phenomenology is involved in each condition is yet to be determined. Here, we aimed to corroborate and expand existing findings, comparing the tendency to experience different types of guilt in different clinical samples. To do so, we presented two clinical samples (i.e., a sample of OCD participants and one of non-OCD participants, composed of patients with anxiety and mood disorders) with diagnostic questionnaires measuring different types of guilt, obsessive-compulsive, and depressive symptoms. 

On the basis of the existing literature, we hypothesize that (i) OC patients would be more prone to experience guilt feelings emerging from the transgression of a moral norm than patients with other disorders; (ii) obsessive traits would be linked to Moral Norm Violation guilt and Moral Dirtiness guilt rather than to Empathy guilt or Harm guilt selectively in OCD patients; and (iii) depression would be linked to Empathy guilt and Harm guilt rather than MNV or MODI guilts.

## 2. Participants

Sixty-eight participants (31 female, 45.59%) were recruited through the “Studio of Psicoterapia Cognitiva” in Rome. Their diagnoses were formulated according to the DSM 5. During a preliminary assessment session, participants were asked to complete a battery of tests, including the MOGS and other measures. The mean age was 34.75 years (SD = 9.9; range = 21–63). Thirty-three participants received a diagnosis of Obsessive Compulsive Disorder (OCD), and the other 35 participants were assigned to the non-OCD group, composed of 11 patients with a diagnosis related to the Anxiety spectrum and 24 participants who received a diagnosis related to Mood disorders. Table 1 displays descriptive statistics for the whole sample. OCD patients had a score on the Yale–Brown Obsessive Compulsive Scale (Y-BOCS) higher than 18.

## 3. Measures

The Moral Orientation Guilt Scale (MOGS) is a 17-item measure that allows the assessment of different types of guilt propensities according to individuals’ moral orientation. The analysis of its latent structure pointed at 4 factors: Moral Norm Violation (MNV), which assesses the fear of having outraged an authority and the attempt to prevent guilt by conforming to moral norms; “Moral Dirtiness” (MODI), measuring the tendency to experience moral disgust towards oneself; “Empathy”, specifically assessing the tendency to feel guilty for the misfortune of others; and “Harm”, measuring the propensity to feel and prevent guilt resulting from harming others. The MOGS has shown good construct validity, and the four subscales and the entire MOGS presented good reliability indices (αMNV = 0.82; αHarm = 0.81; αEmpathy = 0.82; αMODI = 0.70; αTotal = 0.87) [2].

### 3.1. Obsessive-Compulsive Inventory Revised (OCI-R)

Participants were asked to report the extent to which certain everyday life events bothered or stressed them during the past month. The questionnaire was composed of 18 items, and responses were given on a 5-point Likert scale ranging from 1 (not at all) to 5 (extremely). It was composed of 6 factors: Washing (α = 0.60), Obsessing (α = 0.80), Hoarding (α = 0.77), Ordering (α = 0.77), Checking (α = 0.76), and Mental neutralizing (α = 0.61; [30]), each one describing a set of behaviors typical of OCD patients [31].

### 3.2. The Beck Depression Inventory II

The 21-item Beck Depression Inventory-II (BDI-II; [32]) measures the psychological and physical symptoms of depression in adults. Scores range from 0 to 3. The Italian version of the BDI-II has proven to have good internal consistency (alpha = 0.80), as well as good convergent and divergent and criterion validity [33].

### 3.3. Statistical Analysis

To investigate simple differences between the two groups in terms of guilt, we set simple regressions with the group as the unique predictor. Moreover, to test the potential influence between groups across the MOGS scores, we applied a series of multiple regressions, setting the scores of the MOGS subscales and the groups (OCD vs. non-OCD) as predictors. We also considered their interactions. We set the BDI total and subscaless’ scores and the total score of the OCI-R as dependent variables. Note that we performed different models, namely one per response variable. We imputed missing data by using linear regression with a bootstrap method. Whenever we found a statistically significant interaction, we also performed a simple slope analysis. All the models were estimated using the R statistical software [34]. The data imputation was performed using the *MICE* package [35]; the interactions for the bootstrapping procedure were 2000. The effect sizes of each effect were estimated using Cohen’s *d* [36] through the *t_to_d ()* function belonging to the effectsize package [37]. We interpreted the effect sizes following the suggestions of Funder and Ozer (2019) [38]. The simple slope analysis was performed using the *interactions* package [39].

## 4. Results

All the results of the model are shown in Table 2.

Participants within the OCD group obtained higher scores on both Moral Norm Violation (β = 4.42; *p* = <0.01; *d* = 0.84) and Moral Dirtiness (β = 1.84; *p* = 0.03.; *d* = 0.55) subscales, both assessing deontological guilt. No differences were found among the subscales referring to altruistic guilt. Differences between groups in the Moral Norm Violation subscales were also found when associating this subscale with the total score of the OCI-R (β = 1.94; *p* = 0.03; *d* = 0.59): An increase in the Moral Norm Violation score was associated with an increase in the total score of the OCI-R, but only in OC participants (β = 1.38; *p* = <0.001); in the other group, in fact, such a relationship emerged as non-statistically significant (β = −0.56; *p* = 0.24, see Figure 1). Moreover, a main effect of the MODI subscale emerged: A one-unit increase in the MODI score was associated with an increase of 2.64 in the OCI-R’s total score (*p* < 0.001, d = 0.92).

Concerning the BDI-II (Factor 2), we observed a statistically significant interaction between Moral Norm Violation scores and group (β = 1.02; *p* = 0.01; *d* = 0.68). In particular, we observed that, only for participants with a diagnosis of OCD, an increase in the Moral Norm Violation score was associated with an increase in Factor 2 scores (β = 0.57; *p* = 0.04; non-OCD: β = −0.46; *p* = 0.11, see Figure 2). As for the OCI-T total score, a main effect of the MODI subscale emerged: A one-unit increase in the MODI score was associated with an increase of 1.86 in Factor 2 scores (*p* < 0.001, d = 1.07).

A major relationship between Harm and Factor 2 scores emerged, namely an increase in the former scale was associated with an increase in the latter (β = 1.15; *p* = 0.01; *d* = 0.68): Such an effect, in fact, was different across groups (β = −1.87; *p* = 0.01; *d* = −0.67). In particular, we observed, only in participants without a diagnosis of OCD, an increase in Harm scores was associated with an increase in Factor 2 scores (β = 1.15; *p* = < 0.001; OCD group: β = −0.72; *p* = 0.27, see Figure 3).

## 5. Discussion

The results confirm our first and second hypotheses. Specifically, (i) participants within the OCD group obtained higher scores on Moral Norm Violation and Moral Dirtiness subscales, assessing the propensity to experience guilt for disobeying authority (MNV) and the propensity to feel dirty when experiencing guilt and experience self-loathing (MODI). Moreover, (ii) the propensity to feel guilty for having trespassed a moral norm was positively associated with the severity of OCD symptoms selectively in OC patients. These results corroborate behavioral and neural findings, connecting OCD (differently from other disorders) to a type of guilt that is characterized by the infringement of the interiorized moral code rather than the presence of a victim who is suffering [13]. At the behavioral level, OC patients have been shown to be more prone to act in order to respect the “Do not Play God” principle, according to which nobody can take the right to decide who lives and who dies [15], differently from both healthy controls and patients with anxiety disorders. That is the case, for instance, in OC patients’ behavior in the trolley dilemma. The trolley dilemma is a moral problem projecting the participant into an imagined scenario in which a runaway tram or trolley is on course to collide with and kill a number of people (traditionally five) down the track, but a driver or bystander (i.e., the participant) can intervene and divert the vehicle to kill just one person on a different track. In this dilemma, OC patients consistently chose according to deontological rather than consequentialist morality (i.e., they chose to do nothing rather than killing one person in order to save the lives of several others) [16]. Importantly, their preference for the consequentialist choice has previously been shown to be inversely related to the severity of OCD [18]. Our results are also in line with neural data showing that stimuli evoking a type of guilt connected to moral norm violation (such as an angry face presented with sentences like: “How could I behave that immorally!”) are processed differently in OC patients. Specifically, fMRI findings showed that OC patients presented a reduced activation of the ACC, insulae, and putamen in response to these stimuli as compared to control participants. Notably, authors found no significant differences between groups in the processing of stimuli evoking a type of guilt connected with the appraisal of having compromised/disregarded a personal altruistic goal (such as the association of a sad face with a sentence such as: “How unfair! I am doing so well, while she/he is so unlucky!”) [19]. These data have been interpreted as higher cerebral efficiency of OC patients in processing stimuli related to reproach and blame, which might derive from the repeated exposure to these stimuli in OC subjects (i.e., an interpretation consistent with the neuro efficiency hypothesis [40]). The determinants of this facilitated processing are likely ascribable to early learning experiences. Indeed, parental blame and reproach experiences are more present in the memories of obsessive patients than in those of other types of patients [41,42]. Moreover, the family atmosphere is described as being markedly attentive to deontological morality; parental reactions to moral transgressions are often harsh and unpredictable and might imply emotional distance [43]. These experiences might convey to the child a sense of unacceptability as a person and threaten the continuity of the relationship with the caring figure [44].

Our third hypothesis is only partially confirmed. Interestingly, the relationship between depression severity, as assessed by the BDI-II, and guilt propensity seems to be a function of the interaction between the clinical group and guilt type. Specifically, the severity of depressive affective and somatic symptoms in the OCD group is selectively and positively associated with the propensity to experience guilt in reaction to the violation of a moral norm. Conversely, the severity of depressive affective and somatic symptoms in non-OC patients is selectively and positively associated with the propensity to experience guilt in reaction to harming others. In line with this finding, previous studies showed that excessive interpersonal guilt might be a precipitating factor contributing to the emergence of internalizing problems, such as anxiety and depression [26]. Moreover, excessive empathy and hyperaltruistic behaviors have been proposed as risk factors for major depression [28]. Specifically, excessive altruism might lead to maternal depression, contributing to pathogenic guilt in children, which, in turn, creates conditions conducive to risk for developing depression in later adulthood [25]. Furthermore, the association between hyper-altruistic behavior and vulnerability to depression has been supported by neural findings showing an enhanced neural response of the septual/subgenual cingulate cortex, a region consistently implicated in charitable donations, in remitted MDD patients, in line with the hypothesis of a possible association between hyper-altruism and vulnerability to depression [29].

The present study is not exempt from limitations. For instance, the non-OCD sample is composed of both mood-disorder patients and anxiety-disorder patients. Future studies should collect more data on the relationship between guilt and depression, focusing on a sample composed of MDD patients only. Finally, the absence of a control sample of healthy participants limits the possibility of conclusively implicating guilt in the genesis of psychopathology.

Despite these limitations, our results have important clinical implications. Indeed, reducing the propensity to deontological guilt and promoting empathy toward the self might decrease OCD symptoms. Consistently, using Imagery Rescripting to specifically target childhood memories of reproach and blame has been shown to improve symptoms in a single case series study involving OC patients [45].

## Figures and Tables

**Figure 1 jcm-11-04673-f001:**
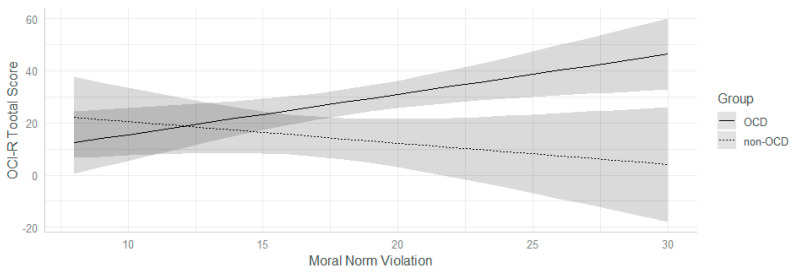
Moral Violation Norm × group interaction on total scores of OCI-R. Grey areas represent confidence intervals. OCI-R Total Score= Obsessive Compulsive Inventory – Revised (Total Score).

**Figure 2 jcm-11-04673-f002:**
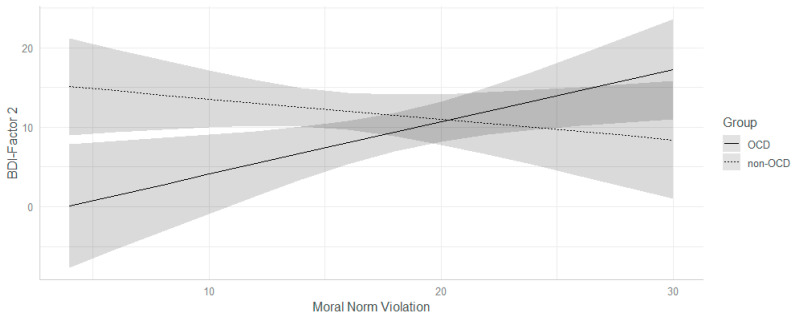
Moral Violation Norm × group interaction on factor 2 scores of BDI. Grey areas represent confidence intervals. BDI= Beck Depression Inventory.

**Figure 3 jcm-11-04673-f003:**
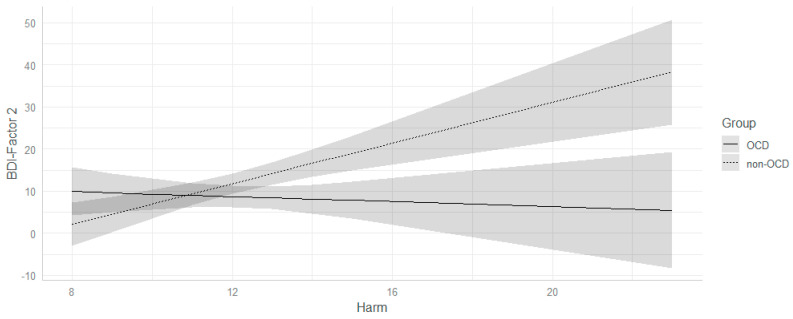
Harm × group interaction on factor 2 scores of BDI. Grey areas represent confidence intervals. BDI= Beck Depression Inventory.

**Table 1 jcm-11-04673-t001:** Descriptives of the used sample. * Note that the percentages of each subsample are estimated starting from the percentage of missingness on the entire sample.

*n* = 68	Frequency (%)	Missingness (%)
Gender		0%
Females Males	45.59% 54.41%	
Diagnosis		
OCD	48.53%	
Non-OCD		
Anxiety spectrum	16.18%	
Mood spectrum	35.29%	
**Measures**	**M (SD)**	
**Moral Orientation Guilt Scale**		
Moral Norm Violation	17.03 (5.73)	0%
*OCD*	19.30 (5.84)	0%
*Non-OCD*	14.89 (4.78)	0%
Empathy	14.07 (3.99)	0%
*OCD*	14.30 (4.21)	0%
*Non-OCD*	13.86 (3.82)	0%
MOral DIrtiness	8.23 (3.49)	0%
*OCD*	9.18 (2.88)	0%
*Non-OCD*	7.34 (3.82)	0%
Harm	12.78 (4.25)	0%
*OCD*	12.72 (2.33)	0%
*Non-OCD*	12.83 (5.52)	0%
**Obsessive-Compulsive Inventory Revised**		
Total score	22.82 (14.98)	27.94%
*OCD*	29.82 (14.96)	26.32% *
*Non-OCD*	13.48 (8.70)	73.68%
**Beck Depression Inventory**		
Factor 1 (cognitive)	10.90 (6.76)	16.18%
*OCD*	11.63 (7.50)	60%
*Non-OCD*	10.26 (6.09)	40%
Factor 2 (affective and somatic)	9.40)	14.71%
*OCD*	10.63 (7.01)	54.54%
*Non-OCD*	8.87 (11.18)	45.46%

**Table 2 jcm-11-04673-t002:** Results of the regression models. Std d stands for standardized Cohen’s d. CI.L/CI.U: Lower and upper 95% confidence intervals.

DV	Predictor	β	Std.Er	*t* Value	P	D	CI.L	CI.U
Moral Norm Violation	Group	4.42	1.29	3.42	**<0.001**	0.84	0.34	1.34
Moral Dirtiness	Group	1.84	0.82	2.23	**0.03**	0.55	0.06	1.04
Empathy	Group	0.45	0.97	0.46	0.65	0.11	−0.37	0.6
Harm	Group	0.45	0.97	0.46	0.65	0.11	−0.37	0.6
OCI-R_Tot	Moral Norm Violation	−0.56	0.47	−1.18	0.24	−0.31	−0.83	0.21
	Group	7.58	15.02	0.5	0.62	0.13	−0.38	0.65
	Moral Dirtiness	2.64	0.75	3.5	**<0.001**	0.92	0.37	1.46
	Empathy	0.86	0.55	1.55	0.13	0.41	−0.11	0.93
	Harm	−1.13	0.56	−2.04	0.06	−0.54	−1.06	−0.01
	Moral Norm Violation: Group	1.94	0.66	2.94	**<0.01**	0.77	0.24	1.3
	Group: Moral Dirtiness	−1.11	1.11	−1	0.32	−0.26	−0.78	0.26
	Group: Empathy	−0.51	0.79	−0.65	0.52	−0.17	−0.68	0.35
	Group: Harm	−1.39	1.21	−1.15	0.26	−0.3	−0.82	0.22
BDI_Factor 1	Moral Norm Violation	0.08	0.3	0.27	0.79	0.07	−0.44	0.59
	Group	−8.32	9.64	−0.86	0.39	−0.23	−0.74	0.29
	Moral Dirtiness	0.49	0.48	1.02	0.31	0.27	−0.25	0.78
	Empathy	−0.03	0.36	−0.08	0.93	−0.02	−0.54	0.49
	Harm	0.15	0.36	0.42	0.68	0.11	−0.41	0.62
	Moral Norm Violation:Group	0.3	0.42	0.71	0.48	0.19	−0.33	0.7
	Group: Moral Dirtiness	−0.02	0.72	−0.02	0.98	−0.01	−0.52	0.51
	Group: Empathy	0.32	0.51	0.63	0.53	0.16	−0.35	0.68
	Group: Harm	−0.21	0.78	−0.27	0.79	−0.07	−0.59	0.44
BDI_Factor 2	Moral Norm Violation	−0.46	0.28	−1.6	0.11	−0.42	−0.94	0.1
	Group	6.71	9.03	0.74	0.46	0.2	−0.32	0.71
	Moral Dirtiness	1.86	0.45	4.09	**<0.001**	1.07	0.52	1.62
	Empathy	0	0.33	−0.01	0.99	0	−0.52	0.51
	Harm	1.15	0.33	3.45	**<0.001**	0.9	0.36	1.44
	Moral Norm Violation: Group	1.02	0.4	2.59	**0.01**	0.68	0.15	1.21
	Group:Moral Dirtiness	−1.27	0.67	−1.9	0.06	−0.5	−1.02	0.03
	Group: Empathy	0.44	0.47	0.92	0.36	0.24	−0.28	0.76
	Group: Harm	−1.87	0.73	−2.56	**0.01**	−0.67	−1.2	−0.14

DV = Dependent Variable; OCI-R_Tot = Obsessive Compulsive Inventory-Revised (Total score); BDI = Beck Depression Inventory. Statistically significant results are presented in bold.

## Data Availability

The datasets used and analyzed during the current study are available from the corresponding author upon reasonable request.

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
