# Peer review of "Guilt Feelings in Obsessive Compulsive Disorder: An Investigation between Diagnostic Groups"

_jcm, 2022, doi:10.3390/jcm11164673_

Round 1

Reviewer 1 Report

The authors present a clinical study comparing guilt among OCD patients with guilt among patients with depressive or anxiety disorders. I have the following specific comments:

1. Abbreviations 'DG' and 'AG' have not been clearly defined

2. The rationale for and importance of differentiating between deontological and altruistic guilt needs to be stronger, i.e. why this study is needed. Why would understanding this improve clinical practice?

3. Grammatical errors need to be checked in the manuscript, e.g. "We condiered also their interaction" in the statistical analysis section.

4. The 4 constructs measured by the guilt scale are unclear. The authors could illustrate their significance with examples from real life examples, perhaps. 

5. Consequently the importance of assessing these in patients is also unclear. 

6. What is the idea behind administering the OCI-R in the non-OCD group? 

7. The question - "whether OC patients are more prone to DG than non-OC patients" is different from asking 'whether OCD is more prone to DG than other disorders'. What the authors have assessed in their study is the latter, since all patients were actively symptomatic. Therefore, the conclusions on the role of DG/AG in the genesis of psychopathology is misleading. 

The manuscript needs major revisions in - a) building a rationale and relevance for this study, and b) interpreting the findings appropriate to the characteristics of the sample studied.

Author Response

  1. Abbreviations 'DG' and 'AG' have not been clearly defined.

Thank you for this comment. We added a clearer definition of “DG and “AG” in the novel version of the manuscript (Lines 34-38):In order to experience deontological guilt (DG) the transgression of an interiorized moral norm is necessary and sufficient, while in order to experience altruistic guilt (AG) the failure of an altruistic goal and the presence of a victim are necessary even if no moral norm has been trespassed1”.

  1. The rationale for and importance of differentiating between deontological and altruistic guilt needs to be stronger, i.e. why this study is needed. Why would understanding this improve clinical practice?

We thank this reviewer for this comment which allows us to better explain the rationale behind the necessity to identify which types of guilt is involved in (which) psychopathology and why this would improve clinical practice. (lines124-162): The central role of guilt in psychopathology is also well represented, for example, inthe analytic tradition, in which guilt is a necessary consequence of individuation 7. In order to rise up above conventional morality and become authentic, one must leave “God, mother and the crowd” 8. Consistently, an overinvestment in the attempt to avoid and prevent the burden of guilt leads to a failure in individuation and to mental disorders. Hence, the suggestion that the confrontation with guilt is the first task of psychological growth (i.e. psychotherapy) (Jung, 1951). Here, we propose that the phenomenology of guilt, involved in different mental disorders, varies according to the type of “original sin” that one is afraid to commit. Under a clinical point of view, understanding which type of guilt the patient is trying to avoid or, using a more cognitivist vocabulary, towards which goal the patient is sinking his (dysfunctional) efforts to escape guilt, is fundamental to build a meaningful treatment plan. Indeed, intervening on types of guilt that deal with moral transgression and with self- moral disgust has different implications than intervening on a more interpersonal type of guilt that focuses on the misfortune of others or the harm caused to another. Namely, the first intervention should address the relationship with moral authority (e.g. God), bringing the patient from a strictly hierarchical conception of it (e.g. the idea that a“sin” is unforgivable and that after the“sin” the relationship with the authority is irreparably compromised) to a more “evangelic” vision where fallibility is intrinsic in human nature and self-forgiveness is possible (Barcaccia and Mancini, 2013). The second type of intervention should assess the causal nexuses between one’s own behavior and the status of the other person, and the construction of an overall balance of guilts and prosocial behaviors and ultimately it should consider intentionality. 

  1. Grammatical errors need to be checked in the manuscript, e.g. "We condiered also their interaction" in the statistical analysis section.

We corrected errors and typos throughout the text.

  1. The 4 constructs measured by the guilt scale are unclear. The authors could illustrate their significance with examples from real life examples, perhaps. 

We now added an appendix containing the 17 MOGS items (translated from the Italian), at the end of the manuscript. Additionally, in the text we reported an Item example for each MOGS subscale and a vignette corresponding to each type of guilt measured by each subscale (lines 42-113).

“The MOGS identifies four different factors: i) Moral Norm Violation – MNV, corresponding to the fear of having outraged an authority and the attempt to prevent guilt by conforming to moral norms (e.g. Item 7: “I feel guilty if do not respect the figures that, to me, represent authority.”).The following vignette well represent a possible real-life scenario in which this type of guilt might be experienced (Lara): “when I was 18-year old I was invited to a party with some of my new friends from university. I would really like to go, because I knew that the boy I liked would have been there, but the party was far from my hometown and my father, who was a very strict man, forbid me to use his old car. I than sneaked out at night and stole my father’s car. I went to the party and parked the car in the street in front of my friend’s house. After a few hours, I decided to go back home but when arrived at the parking spot I noticed that the car had a scratch on one of the side doors. I immediately realized what I had done and pictured the image of my father’s severe expression thinking: “I could have been more careful, He’d be right to be angry”.

  1. ii) Moral Dirtiness – MODI, corresponding to the propensity to feel morally degraded when feeling guilty and to experience moral disgust towards oneself (e.g. Item 1: “When I feel guilty, I feel dirty inside”). An example of this type of guilt is well represented in the following scenario, inspired from real-life account (Paul): When I was 6 my best friend was my cousin Mark who lived next door. We played together all the times and one day, we decided it might be fun to play pretend doctor. To increase the realism, I suggested we should get naked. While we were playing naked, my mother got in the room and asked: “What are these dirty games!?”. I noticed that she was giving me a very disappointed look, almost disgusted, and I felt like I had lost some worth in the eyes of my mother.

iii) Empathy, corresponds to the propensity to feel guilty for the misfortune of others (e.g. Item 17.“I tend to feel guilty if I do not help who is suffering”). This type of guilt is well depicted in the following vignette (Valerie): “One night I went clubbing with a group of friends. We danced all night and had a lot of fun. It was very late when we decided to head back home, walking. Outside it was a chilly winter night and while we were talking, we could see the smoke coming out our mouths. Waiting for a streetlight to become green, I spotted a guy, approximately our age, sitting in the corner of the street, shivering. His clothes were dirty and he had his arms on his head. When the light turned green I hesitated but all my friends kept walking and after a moment I reached them on the other side of the road. The morning after I remembered about that guy in the street. I felt sorry for him and somewhat guilty. I thought that I should have helped him”.   

  1. iv) Harm, corresponds to the propensity to feel and prevent guilt resulting from harming others and the tendency to comfort the victim (i.e. Item 15: If I hurt someone, I feel guilty for the harm I caused”).The following scenario can represent a possible antecedent to this type of guilt. “My brother and I were always rivaling. He is just 1 year younger than me and we always did the same sports. When we were 20 we used to do running races. One day, we were running and I was winning, when I accidentally came too close to him. He tripped and fell and I heard a distinct crack. Once I saw him on the floor bleeding, I realized that I accidently had hurt him. In that moment I started to feel very guilty, he was suffering and, even if I didn’t want to hurt him, it was my fault.”

Even if different types of guilt are often are experienced at the same time in everyday life 3, it is worth noting that in the first two vignettes there is no victim, while in the last two vignettes there is no moral transgression. Similarly, items contained in MNV and MODI subscales are focused on moral transgression and self-loathing with no reference to the concern for others, while in the Empathy and Harm items there is always somebody who is suffering even though no violation of inner moral rule has occurred (see the supplementary materials for MOGS items).”

  1. Consequently, the importance of assessing these in patients is also unclear. 

As it is now pointed out at lines 124-162, to be dysfunctional (pathological) it is not guilt per se. However, the investment that the patient makes in order to escape this emotion quickly becomes dysfunctional and given that guilt is not a unitary concept, the question arises: which type of guilt is the patient trying to avoid? To have a clear representation of the mind of the patient and their goals, could help the clinician to build a more meaningful and effective intervention protocol. Please see also point 2 (in bold).

  1. What is the idea behind administering the OCI-R in the non-OCD group? 

We thank this reviewer for asking this. As now more clearly stated in our hypothesis section (lines 199-296), one of our hypotheses was that obsessive traits would be linked to Moral Norm Violation guilt and MOral DIrtiness guilt rather than to Empathy guilt or Harm selectivelyin OCD patients. This hypothesis is based on the peculiar relation between OCD behaviors and those types of guilt focusing in moral transgression (MNV) and Moral dirtiness/purity (MODI), as already described in the introduction section. Thus, we used a multiple linear regression approach that compared the association between OCI and MOGS scores in the two diagnostic groups and this would have been possible only administering the OCI-R to both clinical samples.

  1. The question - "whether OC patients are more prone to DG than non-OC patients" is different from asking 'whether OCD is more prone to DG than other disorders. What the authors have assessed in their study is the latter, since all patients were actively symptomatic. Therefore, the conclusions on the role of DG/AG in the genesis of psychopathology is misleading. 

Thanks for this comment. What we meant is exactly that “OCD is more prone to the transgression of a moral norm thanother disorders”. In order to clear this out better, we now changed the hypothesis sentence (lines: 199-297): “i) OC patients would be more prone to experience guilt feelings emerging from the transgression of a moral norm than patients with other disorders”.Concerning the role of guilt in OCD, previous studies already compared a sample of healthy control participants with a sample of OCD patients, finding higher guilt scores in the OCD sample. We now reported this study in the text (lines 163-164): The role of guilt in OCD has long been theorized (Freud, 1926) and more recent data confirmed stronger feelings of guilt among OCD patients as compared to matched controls (Gneisser et al., 2019)”.

We now changed our conclusions in the abstract (lines 20-22; 23). Additionally, we added a sentence in the limitation section which now reads: “Finally, the absence of a control sample of healthy participants limits the possibility to conclusively implicate guilt in the genesis of psychopathology.” (lines 525-527).

Reviewer 2 Report

This study investigates an interesting and important element to consider in OCD and depression, which is multidimensional guilt. This investigation has a major strength of being able to compare two clinical samples of similar sizes. However, I have two primary concerns:

One relates to the framing of altruistic guilt (AG) and deontological guilt (DG). There is ample research describing these forms of guilt, though it feels like there is some missing explanation in the introduction regarding what would fall under the two categories for those less familiar with these constructs, as only in the results section is it mentioned that moral dirtiness and moral normal violation are DG and empathy and harm are AG. But more importantly for the design of the study, the guilt scale used by the authors (MOGS) assumes two of its subscales represent AG while two of its other subscales represent DG, but this is not backed by either the factor analysis of the scale conducted in the validation paper nor by any type of factor analysis in the present study. The findings and conclusions regarding AG and DG are further tempered since the two subscales for each form of guilt also are not consistent with one another. To me, unless there is data that can suggest that moral dirtiness and moral norm violation aggregate quantitatively and as would empathy and harm, the authors can only really make conclusions on the subtypes of guilt as assessed by the MOGS rather than comparing AG to DG.

The other concern relates to analyses and presentation of results. If one of the study aims is to compare the two forms of guilt between two different clinical samples, I think it is very important to present means and standard deviations for each subscale for both diagnostic groups. Relatedly, I think it may make more sense from a reader’s perspective to compare these groups using t-tests or an ANOVA as opposed to conducting a regression. Additionally, I think it is important to present findings on all four of the MOGS scales and predictors, even those that are null, perhaps through a table.

Some other comments:

Any particular reason to use two factors for the BDI-II than one, given consistent usage of a single score in research? This could reduce the amount of results that would need to be reported on.

Please be sure to proofread the submission. I noticed a spelling error on line 120 and the use of ‘MMPI’ as opposed to MOGS on line 121.

I also noticed missingness for the BDI and OCI-R scores, but I did not notice any form of multiple imputation or analysis to determine whether data was missing at random. Given the smaller sample size, this seems be important, and it is unclear how or if this was handled.

Overall this was a well-done paper exploring an important content area.

Author Response

REVEWER 2

This study investigates an interesting and important element to consider in OCD and depression, which is multidimensional guilt. This investigation has a major strength of being able to compare two clinical samples of similar sizes. However, I have two primary concerns:

  1. One relates to the framing of altruistic guilt (AG) and deontological guilt (DG). There is ample research describing these forms of guilt, though it feels like there is some missing explanation in the introduction regarding what would fall under the two categories for those less familiar with these constructs, as only in the results section is it mentioned that moral dirtiness and moral normal violation are DG and empathy and harm are AG.

We thank this reviewer for this comment. In the introduction we tried to provide a better definition of each type of guilt measured by the MOGS. We did so also taking into account the second comment of this reviewer (please see trucked changes throughout the manuscript). Thus, the introduction section now contains: a. A more clear definition of AG and DG: In order to experience deontological guilt (DG) the transgression of an interiorized moral norm is necessary and sufficient, while in order to experience altruistic guilt (AG) the failure of an altruistic goal and the presence of a victim are necessary, even if no moral norm has been trespassed1.”b. An example of a MOGS item for each subscale and a vignette portraying an everyday life scenario in which each type of guilts (measured by the MOGS) can be experienced (lines 42-113).“The MOGS identifies four different factors: i) Moral Norm Violation – MNV, corresponding to the fear of having outraged an authority and the attempt to prevent guilt by conforming to moral norms (e.g. Item 7: “I feel guilty if do not respect the figures that, to me, represent authority.”).The following vignette well represent a possible real-life scenario in which this type of guilt might be experienced (Lara): “when I was 18-year old I was invited to a party with some of my new friends from university. I would really like to go, because I knew that the boy I liked would have been there, but the party was far from my hometown and my father, who was a very strict man, forbid me to use his old car. I than sneaked out at night and stole my father’s car. I went to the party and parked the car in the street in front of my friend’s house. After a few hours, I decided to go back home but when arrived at the parking spot I noticed that the car had a scratch on one of the side doors. I immediately realized what I had done and pictured the image of my father’s severe expression thinking: “I could have been more careful, He’d be right to be angry”.

  1. ii) Moral Dirtiness – MODI, corresponding to the propensity to feel morally degraded when feeling guilty and to experience moral disgust towards oneself (e.g. Item 1: “When I feel guilty, I feel dirty inside”). An example of this type of guilt is well represented in the following scenario, inspired from real-life account (Paul): When I was 6 my best friend was my cousin Mark who lived next door. We played together all the times and one day, we decided it might be fun to play pretend doctor. To increase the realism, I suggested we should get naked. While we were playing naked, my mother got in the room and asked: “What are these dirty games!?”. I noticed that she was giving me a very disappointed look, almost disgusted, and I felt like I had lost some worth in the eyes of my mother.

iii) Empathy, corresponds to the propensity to feel guilty for the misfortune of others (e.g. Item 17.“I tend to feel guilty if I do not help who is suffering”). This type of guilt is well depicted in the following vignette (Valerie): “One night I went clubbing with a group of friends. We danced all night and had a lot of fun. It was very late when we decided to head back home, walking. Outside it was a chilly winter night and while we were talking, we could see the smoke coming out our mouths. Waiting for a streetlight to become green, I spotted a guy, approximately our age, sitting in the corner of the street, shivering. His clothes were dirty and he had his arms on his head. When the light turned green I hesitated but all my friends kept walking and after a moment I reached them on the other side of the road. The morning after I remembered about that guy in the street. I felt sorry for him and somewhat guilty. I thought that I should have helped him”.   

  1. iv) Harm, corresponds to the propensity to feel and prevent guilt resulting from harming others and the tendency to comfort the victim (i.e. Item 15: If I hurt someone, I feel guilty for the harm I caused”).The following scenario can represent a possible antecedent to this type of guilt. “My brother and I were always rivaling. He is just 1 year younger than me and we always did the same sports. When we were 20 we used to do running races. One day, we were running and I was winning, when I accidentally came too close to him. He tripped and fell and I heard a distinct crack. Once I saw him on the floor bleeding, I realized that I accidently had hurt him. In that moment I started to feel very guilty, he was suffering and, even if I didn’t want to hurt him, it was my fault.”
  2. A sentence that clears out that: “Even if different types of guilt are often are experienced at the same time in everyday life 3, it is worth noting that in the first two vignettes there is no victim, while in the last two vignettes there is no moral transgression. Similarly, items contained in MNV and MODI subscales are focused on moral transgression and self-loathing with no reference to the concern for others, while in the Empathy and Harm items there is always somebody who is suffering even though no violation of inner moral rule has occurred (see the supplementary materials for MOGS items).”(lines 114-120).

  1. But more importantly for the design of the study, the guilt scale used by the authors (MOGS) assumes two of its subscales represent AG while two of its other subscales represent DG, but this is not backed by either the factor analysis of the scale conducted in the validation paper nor by any type of factor analysis in the present study. The findings and conclusions regarding AG and DG are further tempered since the two subscales for each form of guilt also are not consistent with one another. To me, unless there is data that can suggest that moral dirtiness and moral norm violation aggregate quantitatively and as would empathy and harm, the authors can only really make conclusions on the subtypes of guilt as assessed by the MOGS rather than comparing AG to DG.

We thank this reviewer for this comment. In following, in the discussion section we now tempered our conclusions drawing them specifically on the subtypes of guilt assessed by the MOGS (please see trucked changes throughout the discussion section), e.g.:

 “Results confirm our first and second hypotheses. Specifically: i) participants within the OCD group obtained higher scores on Moral Norm Violation and MOral DIrtiness subscales, assessing the propensity to experience guilt for disobeying authority (MNV) and the propensity to feel dirty when experiencing guilt and to experience self-loathing (MODI). Moreover, ii) the propensity to feel guilty for having trespassed a moral normwas positively associated with the severity of OCD symptoms selectively in OC patients. These results corroborate behavioral and neural findings, connecting OCD (differently from other disorders) to a type of guilt that is characterized by the infringement of the interiorized moral code rather than the presence of a victim who is suffering9.”Lines 429-438.

  1. The other concern relates to analyses and presentation of results. If one of the study aims is to compare the two forms of guilt between two different clinical samples, I think it is very important to present means and standard deviations for each subscale for both diagnostic groups.

We thank the Reviewer for giving us the chance of being more precise. Means and standard deviations of each group has been added inside the revised version of the paper.

  1. Relatedly, I think it may make more sense from a reader’s perspective to compare these groups using t-tests or an ANOVA as opposed to conducting a regression.

We agree with the Reviewer that readers often read about ANOVA and t-test. Nonetheless, we respectfully disagree about the fact that comparing means using t-test or ANOVA makes more sense than conducting a regression, for a series of reasons: (a) t-test is a special case of ANOVA that, in turn, is a special case of linear regression. Therefore, the estimation of t-test ad ANOVAs with our data would conduct to identical results. (b) ANOVA is advisable when all the predictors are categorical variables, otherwise ANCOVA would be preferable. ANCOVA is a special case of multiple regression where predictors are both continuous and categorical, with the difference that multiple regression is less biases than ANCOVA; some of the models suggested by our work deals with different kind of predictors. Therefore, it remains preferable using (multiple) regressions. (c) Finally, linear regression provides a standard framework for reading a series of models, since for each effect is described by a beta coefficient (representing the difference among the means of the conditions). This brings coherence while reading the results, since all the models are written using the same format. Differently, the use of t-tests and ANOVA would require the use of different statistics (i.e., student’s t or F/) that do not have the same meanings. For instance, some authors pointed out that an F value tells the reader that an effect came out, but not which one and/or its direction. A beta coefficient retrieved from a regression output provides, actually, all these data. Therefore, it is a clearer and more precise way to compare group means. We report some references supporting all the points that we provided.

Eisenhauer, J. G. (2006). How a dummy replaces a student's test and gets an F (or, how regression substitutes for t tests and ANOVA). Teaching Statistics28(3), 78-80.

Nelson, L. R., & Zaichkowsky, L. D. (1979). A case for using multiple regression instead of ANOVA in educational research. The Journal of Experimental Education47(4), 324-330.

Platt, R. W. (1998). ANOVA, t tests, and linear regression. Injury prevention4(1), 52-53.

Schad, D. J., Vasishth, S., Hohenstein, S., & Kliegl, R. (2020). How to capitalize on a priori contrasts in linear (mixed) models: A tutorial. Journal of Memory and Language110, 104038.

  1. Additionally, I think it is important to present findings on all four of the MOGS scales and predictors, even those that are null, perhaps through a table.

As suggested by the Reviewer, we presented all the outputs of the linear models, formatted as a table. In particular, we updated and shifted the Table contained in Appendix into the main document.

  1. Some other comments:

Any particular reason to use two factors for the BDI-II than one, given consistent usage of a single score in research? This could reduce the amount of results that would need to be reported on.

We thank this reviewer for this comment. It might be that the routine in research is to use a single score for BDI but research also consistently showed that the BDI latent structure is composed by two factors, sometimes three (Shafer, 2006). Item loadings vary a little across studies but here we used Beck’s 2-factor solution (Beck et al.,1996). Moreover,“a study which meta-analyzed 51 studies comprising 62 samples (N = 20,475) providing pattern matrices, determined that the two-factor solution comprising Cognitive and Somatic-Affective factors was supported for the full sample. The two-factor solution was also supported for subgroups of studies.”The same study found the existence of a general depression factor, which was supported by the good fit of the one-factor model. However…: “the finding of a general depression factor does not necessarily invalidate the differentiation of specific depression factors as a different depression subfactor was reportedly related to different etiologies and external criteria (Abramson, Metalsky, & Alloy, 1989; Alloy, Just, & Panzarella, 1997; Payne, Palmer, & Joffe, 2009)”

Huang, C., & Chen, J.-H. (2015). Meta-Analysis of the Factor Structures of the Beck Depression Inventory–II. Assessment22(4), 459–472. https://doi.org/10.1177/1073191114548873

Please be sure to proofread the submission. I noticed a spelling error on line 120 and the use of ‘MMPI’ as opposed to MOGS on line 121.

We proofread errors and typos throughout the text.

I also noticed missingness for the BDI and OCI-R scores, but I did not notice any form of multiple imputation or analysis to determine whether data was missing at random. Given the smaller sample size, this seems be important, and it is unclear how or if this was handled.

We thank the Reviewer for giving us the opportunity to implement this aspect. We imputed missing data, by using the R package MICE. As imputation method, we used the “Linear regression using bootstrap” one, with 2000 iterations. We added this specification also inside the new version of the manuscript. All the previous results were confirmed. Moreover, this change led to additional effects, that we reported and discussed.

Overall this was a well-done paper exploring an important content area.